

# BERT-5mC: an interpretable model for predicting 5-methylcytosine sites of DNA based on BERT

Shuyu Wang, Yinbo Liu, Yufeng Liu, Yong Zhang and Xiaolei Zhu

School of Sciences, Anhui Agricultural University, Hefei, Anhui, China

## ABSTRACT

DNA 5-methylcytosine (5mC) is widely present in multicellular eukaryotes, which plays important roles in various developmental and physiological processes and a wide range of human diseases. Thus, it is essential to accurately detect the 5mC sites. Although current sequencing technologies can map genome-wide 5mC sites, these experimental methods are both costly and time-consuming. To achieve a fast and accurate prediction of 5mC sites, we propose a new computational approach, BERT-5mC. First, we pre-trained a domain-specific BERT (bidirectional encoder representations from transformers) model by using human promoter sequences as language *corpus*. BERT is a deep two-way language representation model based on Transformer. Second, we fine-tuned the domain-specific BERT model based on the 5mC training dataset to build the model. The cross-validation results show that our model achieves an AUROC of 0.966 which is higher than other state-of-the-art methods such as iPromoter-5mC, 5mC_Pred, and BiLSTM-5mC. Furthermore, our model was evaluated on the independent test set, which shows that our model achieves an AUROC of 0.966 that is also higher than other state-of-the-art methods. Moreover, we analyzed the attention weights generated by BERT to identify a number of nucleotide distributions that are closely associated with 5mC modifications. To facilitate the use of our model, we built a webserver which can be freely accessed at: http://5mc-pred.zhulab.org.cn.

## INTRODUCTION

DNA 5-methylcytosine (5mC) is one of the most extensively studied epigenetic modifications in eukaryotes (*de Mendoza, Lister & Bogdanovic, 2020*), which is the product of covalently adding a methyl group to the fifth atom of the cytosine pyrimidine ring (*Arand et al., 2012*). It has been demonstrated that 5mC is involved in a range of developmental and physiological processes in multicellular eukaryotes, such as gene expression, genome maintenance, parental imprinting, and transcriptional regulation.

5mC is most commonly found in the CpG dinucleotide environment, whose dysfunction is associated with many diseases. The 5mC level of promoters has been linked to immune system regulation. The abnormal methylation of cytosine can lead to

Corresponding authors
Yong Zhang,
yongzhang@ahau.edu.cn
Xiaolei Zhu,
xlzhu_mdl@hotmail.com

autoimmune rheumatic diseases such as systemic lupus erythematosus (*Javierre et al., 2010*) and rheumatoid arthritis (*Rodríguez-Ubreva et al., 2019*). Recently, *Kaur et al. (2022)* reviewed how the 5mC level relates to Alzheimer's disease and other neurodegenerative disorders. Moreover, altered levels of DNA methylation are a major cause of cancer, for example, the hypomethylation and local hypermethylation of the promoters of tumor suppressor genes play a key role in carcinogenesis (*Agrawal, Murphy & Agrawal, 2007*; *Frigola et al., 2005*). It has been reported that epigenetic modifications have various impacts on disease development, and DNA methylation is often used as a clinical biomarker and a key tool for disease diagnosis and treatment due to its high stability and physical association with specific DNA sequence patterns (*Ballestar, Sawalha & Lu, 2020*). The accurate identification of 5mC sites of promoters across the genome is critical for understanding the mechanisms of DNA methylation in human diseases, and it also provides important clues for disease treatment.

Currently, experimental assays for identifying 5mC sites with single base resolution can be classified into three categories, namely chemical conversion-based, enzyme-based, and directly DNA sequence-based methods (*Lv et al., 2021*). As a kind of chemical conversion-based method, bisulfite sequencing (BS-seq) (*Frommer et al., 1992*) is considered the gold standard technology for 5mC detection. Based on the fragments randomly generated from the genome, the BS-seq can perform the whole-genome bisulfite sequencing (WGBS) assay for different species (*Stevens et al., 2013*). Considering the limitations of traditional BS-seq based WGBS, *Meissner (2005)* developed the reduced representation bisulfite sequencing (RRBS) and *Miura et al. (2012)* developed the post-bisulfite adaptor tagging (PBAT) method. Although these experimental methods have the capability to map 5mC sites in genome-wide with single-base resolution, the time and economic costs required to carry out bio-sequencing experiments are often prohibitive in the post-genomic era. Consequently, it is imperative to explore efficient computational methods to detect 5mC sites.

Recently, several computational methods have been developed to predict DNA 5mC sites. *Bhasin et al. (2005)* developed a model called Methylator to predict 5mC sites in CpG dinucleotides. In their method, the traditional binary sparse representation was utilized to encode the nucleotides of DNA sequences and support vector machines (SVM) was used to train the model. Similarly, *Liu et al. (2015)* constructed a predictor called iDNA-Methyl to predict methylation sites. In the method, the pseudo-nucleotide composition (pseTNC) was used to encode the DNA sequence (*Liu et al., 2015*), and SVM was used as the classifier. *Zhang, Xiao & Xu (2020)* developed a predictor, iPromoter-5mC, to predict the 5mC sites of the promoters by using a deep neural network (DNN), considering the relationship between the 5mC level of promoters and the proliferation of cancer cells. Based on the same benchmark datasets as that used for building iPromoter-5mC, *Nguyen et al. (2021)* extracted k-mer embeddings from DNA sequences by using a natural language model called FastText (*Bojanowski et al., 2017*; *Joulin et al., 2016*) and then developed their model based on XGBoost. Recently, *Cheng et al. (2021)* developed a model, BiLSTM-5mC, to predict 5mC sites of promoters based on the BiLSTM network. Based on the same data sets, *Jia, Qin & Lei (2023)* proposed a model called DGA-5mC to predict 5mC sites of

promoters. In their method, three feature encoding methods, one-hot, nucleotide chemical property coding (NCP) and nucleotide density coding (ND), were used to extract features from DNA sequences and the improved densely-connected convolutional network (DenseNet) and bi-directional GRU network were used to train the model. In addition to the methods that were specifically developed for predicting the 5mC site, there are computational methods (*Lv et al., 2020*; *Pavlovic et al., 2017*; *Yu et al., 2021*) that can predict multiple methylation sites for multiple species. Recently, *Jin et al. (2022)* developed a model, iDNA-ABF, to predict 6mA, 4mC, 5mC and 5hmC sites of multiple species, which is based on a multiple-scale DNA language model, DNABERT (*Ji et al., 2021*). DNABERT is a pre-trained model based on bidirectional encoder representations from transformers (BERT) which is a popular natural language processing model proposed by *Devlin et al. (2018)*. More specific, BERT is a deep two-way language representation model based on Transformer. In addition, *Zeng, Gautam & Huson (2023)* built a model, MuLan-Methyl, by integrating the predicted results of five popular transformer-based language models. These computational models indicate that the DNA sequence contains the relevant information for identifying 5mC sites.

As mentioned above, great progress has been made in predicting DNA 5mC sites, and BERT has been transferred for predicting DNA methylation sites, which achieved promising results. The BERT model was also used in other biological sequence-based tasks and has achieved superior performance (*Ho, Le & Ou, 2021*; *Le et al., 2021*). There are two strategies to apply pre-trained language models for downstream tasks: feature-based and fine-tuning. As an example of feature-based approaches, *Qiao, Zhu & Gong (2022)* developed a model called BERT-Kcr to predict the Kcr sites of proteins. In their method, the features encoded by BERT were extracted and fed to the BiLSTM network for training. As an example of fine-tuning methods, *Zhang et al. (2021a)* used six different AMP datasets to fine-tune a protein-language-based BERT model and finally constructed a general AMP recognition model to realize the accurate recognition of antimicrobial peptides. Although general biological sequence BERT models have been used to predict 5mC sites, the domain-specific model might be able to improve the performance.

In this work, we pre-trained a domain-specific BERT model to construct a new computational method (BERT-5mC) to predict 5mC sites. The workflow is illustrated in Fig. 1. First, we collected promoter sequences from UCSC (http://genome.ucsc.edu/) for pre-training the promoter-specific BERT model. Subsequently, the pre-trained model was fine-tuned to predict 5mC sites by using the training dataset. The performance on the independent test set indicates that our model, BERT-5mC, achieves an area under the receiver operating characteristic curve (AUROC) of 0.966, which is superior to other available predictors.

## MATERIALS AND METHODS

### Benchmark datasets

To ensure the data reliability and fairness in prediction performance comparison, we used the datasets collected by *Zhang, Xiao & Xu (2020)*. Prior to the advent of Zhang's dataset, several other datasets had been collected for building the corresponding 5mC predictors
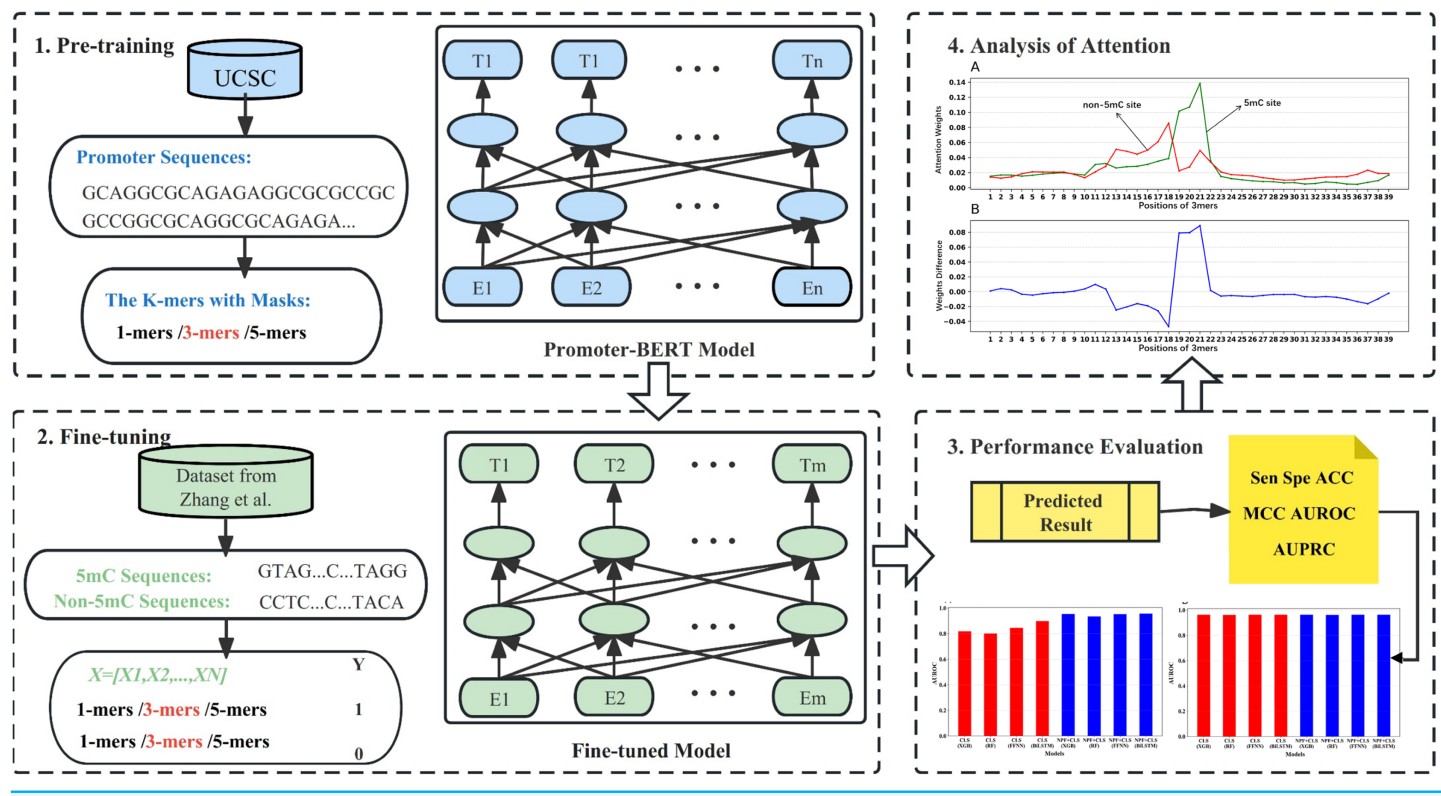

**Figure 1 The overall flow of analysis in the present study.**

(*Bhasin et al., 2005*; *Fang et al., 2006*; *Liu et al., 2015*); however, these datasets have limitations in small sample size and redundancy, which can lead to overestimation of the models. Based on the promoter 5mC position information of small cell carcinoma (SCLC), *Zhang, Xiao & Xu (2020)* built a comprehensive and non-redundant dataset. Specifically, they collected the 5mC position information from the cancer cell line encyclopedia (CCLE) database (*Barretina et al., 2012*; *Li et al., 2019*), and then downloaded the promoter sequences from the UCSC genome browser (http://genome.ucsc.edu/). Based on the 5mC position information, a total of 66,094 original promoter sequences with a length of 1,041 bp were obtained. Each sequence was then segmented into multiple 41 bp fragments with cytosine in the middle. To eliminate the redundancy, the CD-HIT program was used to remove similar segments by using a sequence similarity threshold of 80% (*Fu et al., 2012*). The final dataset contains a total of 69,750 positive samples and 823,576 negative samples. To validate and test the predictive models, the dataset was divided into a training dataset and an independent test set. Specifically, 80% of the data were randomly selected from the positive and negative samples, respectively, as the training set (55,800 5mC samples and 658,861 non-5mC samples), and the remaining 20% of the data were used as the independent test set (13,950 5mC samples and 164,715 non-5mC samples). These datasets have been utilized in several recent works of *Nguyen et al. (2021)*, *Cheng et al. (2021)*, *Jia, Qin & Lei (2023)*. The datasets are available at https://github.com/sgrwang/BERT-5mC. Details of the datasets are shown in Table 1.

**Table 1 The information of the benchmark datasets.**

| Dataset | Positive sample | Negative sample |
| --- | --- | --- |
| Training dataset | 55,800 | 658,861 |
| Testing dataset | 13,950 | 164,715 |
| Total | 69,750 | 823,576 |

## Pre-training the promoter-BERT model

In some sense, the DNA sequences are similar to natural languages. DNA sequences consist of the letters from a chemically defined alphabet, nucleotides. Similar to natural languages, these letters can form secondary structural elements ("words"), which assemble to form domains ("sentences") that undertake a function ("meaning"). Thus, it is reasonable to transfer natural language models to process DNA sequences. Bidirectional Encoder Representations from Transformers (BERT), developed by *Devlin et al. (2018)*, comprising of multiple encoder layers. Each encoder layer is composed of two sub-layers: multi-head self-attention and feed-forward neural network. The multi-head self-attention sub-layer, based on the scaling point product attention (self-attention mechanism), is the core component, and is capable of parallel computing with a strong capacity to capture word dependencies within sentences.

Considering the certain differences between DNA sequences and natural languages, we pre-trained a domain-specific BERT model based on the *corpus* of human promoter sequences. The pre-training data were downloaded from UCSC (http://genome.ucsc.edu/), and the sequences of 1,041 bp region located upstream of the Transcription start sites (TSS) were taken as the promoter sequence. The constructed pre-training *corpus* contains 1,825,095 promoter sequences. Each sequence was split into a series of k-mers (overlapping is allowed) as words of the sentence. By referring to the work of DNABERT and considering the large size of our pre-training *corpus* and the computational resource of our lab, three kinds of k-mers with k equaling 1, 3, or 5 were used as words, resulting in three pre-trained DNA language BERT models. We pre-trained our DNA language BERT models based on BERT-base, which consists of 12 encoder layers. The models were trained for 200,000 epochs with a learning rate of 2E−5. Subsequently, we fine-tuned and evaluated the three Promoter-BERT models based on the datasets used in this study.

## Feature encoding methods

### Extraction of BERT embeddings

The embeddings of DNA segments can be extracted from a pre-trained natural language BERT model or a pre-trained domain-specific BERT model. From the BERT model, each word of an input sentence can be encoded as a 768-dimensional embedding vector by each encoder layer. However, due to the large amount of data in this study, with the training set comprising 55,800 positive samples and 658,861 negative samples, only the embedding of the "CLS" token was utilized as features. As stated in *Devlin et al. (2018)*: "the first token of every sequence is always a special classification token ([CLS]). The final hidden state

corresponding to this token is used as the aggregate sequence representation for classification tasks". Note that we extracted the embeddings from the last encoder layer of the BERT model.

### Other feature encoding methods

The nucleotide nature and frequency (NPF) contain sequence order and position-specific information and have been widely employed in the prediction of DNA or RNA modification sites (*Xia et al., 2019*; *Xu et al., 2019*). According to the definition of NPF, each nucleotide $n_i$ ($1 \leq i \leq 41$) in the DNA sequence is represented as a 4-dimensional vector ($x_i, y_i, z_i, d_i$). $x_i$, $y_i$, and $z_i$ represent the ring structure, chemical functional groups, and hydrogen bonds of nucleotide $n_i$, respectively, with values of 0 or 1. $d_i$ is the cumulative frequency of nucleotide $n_i$ in the prefix string. The exact calculation process is as follows.

$$x_i = \begin{cases} 1, \text{if } n_i \in \{A, G\} \\ 0, \text{otherwise} \end{cases}$$

$$y_i = \begin{cases} 1, \text{if } n_i \in \{A, C\} \\ 0, \text{otherwise} \end{cases}$$

$$z_i = \begin{cases} 1, \text{if } n_i \in \{A, T\} \\ 0, \text{otherwise} \end{cases}$$

$$d_i = \frac{1}{|N_i|} \sum_{j=1}^{|N_i|} f(n_j)$$

where

$$f(n_j) = \begin{cases} 1, \text{if } n_j = n_i \\ 0, \text{otherwise} \end{cases}, n_i, n_j \in \{A, C, G, T\}.$$

## Learning algorithms

### Extreme gradient boosting (XGBoost)

As a kind of ensemble learning algorithm, XGBoost (*Chen & Guestrin, 2016*) has gained much attention for its excellent performance and efficient training speed. It improves on the gradient boosting decision tree (GBDT) by, for example, extending the loss function from a squared loss to a second-order derivable loss, adding regularization terms, *etc*. As a forward addition model, the core of XGBoost includes the boosting algorithm, in which multiple weak learners are integrated into a strong learner.

### Random forest (RF)

Random forest (RF) (*Breiman, 2001*) is also a kind of ensemble learning algorithm that uses decision trees as the base learners. The "randomness" of the random forest comes from two strategies. Firstly, the training subset for building each base learner is obtained by using a bootstrap-resampling technique. Secondly, the feature subset used for constructing each base learner is randomly selected from all the features. Then, the majority voting

strategy is used to determine the predicted label of a test sample. In essence, it is an improvement to the decision tree algorithm, increasing the randomness in building base learners and thus reducing the variance of the final model (*Oshiro, Perez & Baranauskas, 2012*).

### Feed-forward neural network (FFNN)

A feed-forward neural network (FFNN) is a unidirectional multilayer network where information is passed unidirectionally, layer by layer, starting from the input layer and ending at the output layer. A feed-forward neural network consists of three parts: the input layer, the output layer, and the hidden layers (which can be one or more layers) (*Hemeida et al., 2020*). Each layer contains different nodes. The numbers of nodes for the input layer and the output layer are determined based on the dimension of the input feature and the dimension of the target variable. However, the number of nodes in a hidden layer is changeable. The nodes in each layer are fully connected to the nodes of its neighboring layers, but there are no in-layer or cross-layer connections between nodes.

### Bidirectional long short-term memory (BiLSTM)

To solve the gradient disappearance problem of recurrent neural networks (RNNs), *Hochreiter & Schmidhuber (1997)* proposed the long short-term memory (LSTM) network, which has been widely used in bioinformatics and achieved good performance (*Abbasi et al., 2020*; *Kumar et al., 2021*; *Tsukiyama et al., 2021*; *Zhang et al., 2021b*). However, standard LSTM units can only focus on historical information and ignore future information when processing sequence data. BiLSTM consists of forward and backward LSTM neural networks, so that both historical and future information can be captured (*Graves, Fernández & Schmidhuber, 2005*).

## Evaluation metrics

To measure the generalization of the models fairly and comprehensively, we performed 5-fold cross-validation and independent dataset testing based on the benchmark datasets. Four commonly used evaluation metrics, namely, sensitivity (Sen), specificity (Spe), accuracy (ACC), and Matthew's correlation coefficient (MCC), were used to evaluate our models, which can be calculated by the following formulas:

$$\text{Sen} = \frac{TP}{TP + TN},$$

$$\text{Spe} = \frac{TN}{TN + FP},$$

$$\text{ACC} = \frac{TP + TN}{TP + TN + FP + FN},$$

$$\text{MCC} = \frac{TP \times TN - FP \times FN}{\sqrt{(TP + FP) \times (TP + FN) \times (TN + FP) \times (TN + FN)}},$$

where TN, TP, FN, and FP represent the number of true negatives (negative samples that are correctly predicted), true positives (positive samples that are correctly predicted), false negatives (positive samples that are wrongly predicted to be negative samples), and false

positives (negative samples that were incorrectly predicted as positive samples), respectively. Sen and Spe are the predictive accuracies of the model to identify 5mC sites and non-5mC sites, respectively. ACC is the predictive accuracy for all the samples. The range of these three values is [0, 1]. The larger the value, the more accurate the prediction of the model. The ACC value gives us an intuitive ratio for the model performance. However, in the case of unbalanced data sets, MCC is more reasonable than ACC for evaluating the overall performance of the model. To obtain a high MCC value, the model needs to make correct predictions on both positive and negative samples, independently of their ratio in the dataset (*Chicco & Jurman, 2020*). Furthermore, to make the experimental results more comprehensive, we plotted receiver operating characteristic curves (ROC) and precision-recall curves (PRC), and calculated the area under ROC (AUROC) and the area under PRC (AUPRC) as evaluation metrics for this study. Although MCC, ACC can be used to evaluate the overall performance of the model, their values depend on the selection of the decision threshold. The ROC and PRC curves and the areas under the curves do not depend on the selection of the decision threshold. Notably, all experimental results are interpreted using a decision threshold of 0.3. In principle, the threshold reflects the ratio between positive samples and negative samples. In our case, the ratio between positive and negative samples in the training dataset is about 0.08, we have tried several different decision thresholds such as 0.1, 0.2, 0.3, 0.4, and the optimal ACC and MCC values can be obtained when 0.3 is selected.

## RESULTS

### Word size selection

Since we considered the DNA segments as sentences for the natural language models, the k-mers of the DNA segments were used as words for the sentences. The size of k-mers (or words) could affect the performance of the BERT model in identifying 5mC sites, thus necessitating the selection of an optimal k value of k-mers. We first pre-trained three BERT models with word sizes of 1-mers, 3-mers, and 5-mers, respectively, and then fine-tuned the three models based on the samples of the training set to evaluate the representability of the three models.

As illustrated in Fig. 2 and Table 2, the model constructed with k = 3 generally outperforms the models constructed with k = 1 or k = 5. When the value of k is 3, the model has higher AUROC, AUPRC, MCC, and ACC than the other two models for the five-fold cross-validation on the training set. Specifically, the AUROC value is 0.966, which is higher than 0.959 (k = 1) and 0.964 (k = 5), and the AUPRC value is 0.602, which is higher than 0.554 (k = 1) and 0.575 (k = 5). This suggests that the BERT model pre-trained on 3-mers has the best performance in identifying the 5mC sites. Note that the three fine-tuned models were also evaluated on the independent test set. As shown in Table S1, the model based on 3-mers is again superior to the other two models. Therefore, we set k to 3 for the following experiments. Note that although our results show that the best performance is obtained when k equals 3, it is possible that better results could be obtained when 2-mer or 4-mer is used to pre-train and fine-tune the model.
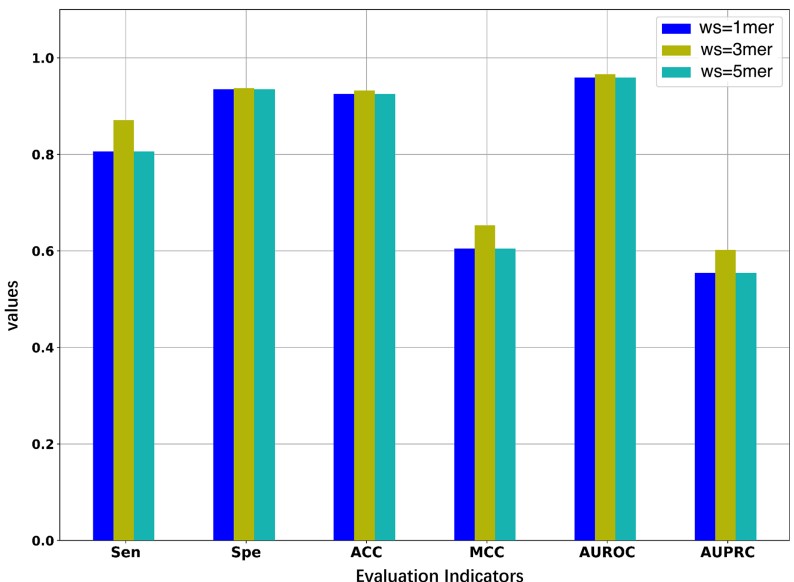

**Figure 2 Performance comparison of models under different word sizes (ws) based on the 5-fold cross-validation of the training dataset.**

**Table 2 Performance evaluation of models based on different window sizes by using the 5-fold cross-validation on the training dataset.**

| Word sizes | Sen | Spe | ACC | MCC | AUROC | AUPRC |
|---|---|---|---|---|---|---|
| 1mer | 0.806 | 0.935 | 0.925 | 0.605 | 0.959 | 0.554 |
| 3mer | 0.871 | 0.937 | 0.932 | 0.653 | 0.966 | 0.602 |
| 5mer | 0.836 | 0.940 | 0.932 | 0.640 | 0.964 | 0.575 |

## Models based on BERT embedding features

Apart from fine-tuning, features can be extracted from BERT to solve the downstream tasks. In this study, the embedding of the token "CLS" was extracted as features from the last encoder layer of the pre-trained Promoter-BERT model (trained on the promoter sequences downloaded from UCSC). Additionally, we also extracted the embedding of the token "CLS" as features from the last encoder layer of the fine-tuned Promoter-BERT model (fine-tuned using the training dataset). Note that the fine-tuned BERT models were based on five-fold cross-validation to avoid information leakage. Besides, we also extracted NPF features and combined them with the previous two BERT embedding features as the third and fourth feature sets, respectively. Based on the four feature sets, four classifiers including XGBoost, RF, BiLSTM, and FFN, were used to train the models, resulting in a total of 16 models.

The five-fold cross-validation was conducted for the 16 models on the training set. Figure 3 shows the average AUROC values of the cross validation for the 16 models. We optimized the hyperparameters of the four learning algorithms; for more information, see Tables S2–S4. As illustrated in Fig. 3A, for the embedding features extracted from the
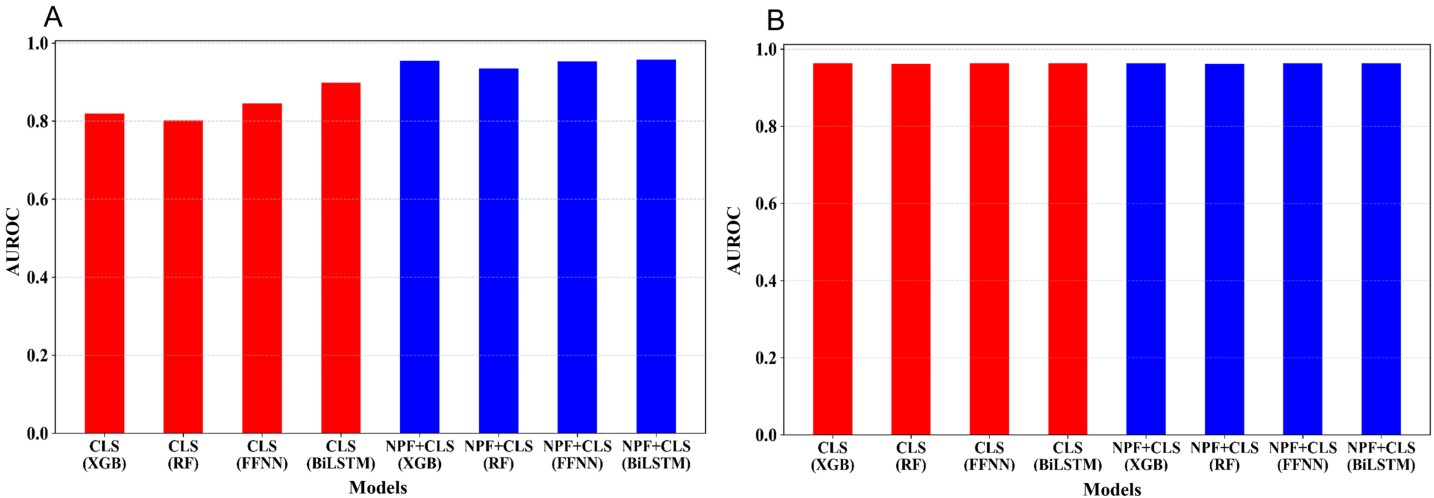

**Figure 3 Performance comparison between models based on different feature combinations and different learning algorithms.** (A) Cross-validation AUROCs of the models based on the embedding features extracted from the Promoter-BERT model. (B) Cross validation AUROCs of the models based on the embedding features extracted from the fine-tuned model.

pre-trained Promoter-BERT model, adding NPF is helpful for improving the model's performance. For example, to the models built by XGBoost, the AUROC based on the Promoter-BERT embedding features alone is only 0.819; however, with the addition of the NPF features, the AUROC of the model increases to 0.955. Thus, an improvement of 0.142 is achieved, which indicates that the sequence information contained in the embedding features of the pre-trained Promoter-BERT model can be complemented by NPF features. Furthermore, we conducted experiments to extract embeddings from the fine-tuned Promoter-BERT model. As depicted in Fig. 3B, the four models based on the embedding features extracted from the fine-tuned Promoter-BERT model show good cross-validation performance, and adding the NPF feature cannot improve the performance. These results indicate that the representability of the embeddings extracted from the fine-tuned Promoter-BERT model is better than that of those extracted from the pre-trained Promoter-BERT model. Specifically, as shown in Fig. 3B and Table S3, the AUROC values are 0.963, 0.962, 0.964, and 0.964 for the models built by using XGBoost, RF, FFNN, and BiLSTM based on the embedding extracted from the fine-tuned Promoter-BERT model, respectively, which are higher than the AUROC values based on the embedding extracted from the pre-trained Promoter-BERT model.

Moreover, the average embeddings of the 39 3-mers of the samples were also extracted from the fine-tuned Promoter-BERT to build the models. As shown in Table S5, the performance of these models is similar to that of the models based on the embedding of 'CLS' (Table S3).

## Comparison between fine-tuned model and the model based on embedding features extracted from Promoter-BERT

In the above two sections, we used two strategies to construct the 5mC site prediction models: one was to fine-tune the pre-trained Promoter-BERT model end-to-end, and the

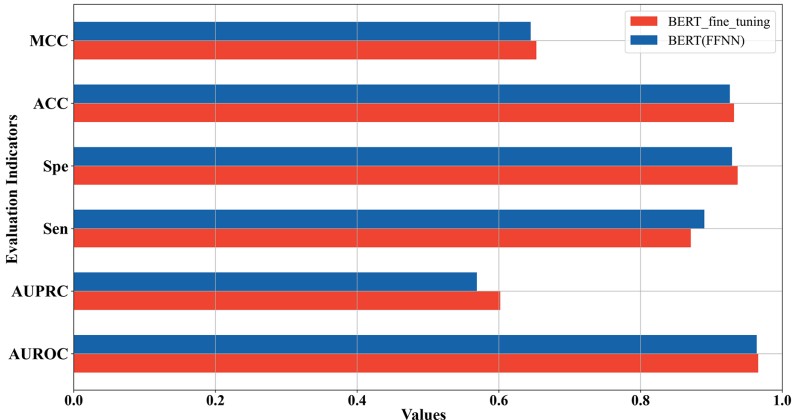

**Figure 4** Performance comparison between the optimal model based on end-to-end fine-tuning and the optimal model based on embedding features extracted from Promoter-BERT.

**Table 3** Performance comparison between different models on the training dataset by using the 5-fold cross validation. The highest value in each column is shown in bold.

| Method | Sen | Spe | ACC | MCC | AUROC |
|---|---|---|---|---|---|
| iPromoter-5mC | 0.875 | 0.904 | 0.902 | 0.574 | 0.957 |
| 5mC_Pred | 0.899 | 0.920 | 0.918 | 0.626 | 0.962 |
| BiLSTM-5mC | 0.810 | **0.940** | 0.930 | 0.624 | 0.964 |
| DGA-5mC | **0.904** | 0.925 | 0.923 | 0.642 | 0.964 |
| BERT-5mC | 0.871 | 0.937 | **0.932** | **0.653** | **0.966** |

other was to extract the embeddings of the last encoder layer of the fine-tuned Promoter-BERT model as features to train the model. By the first strategy, the model performs best when the value of k is 3. By the second strategy, the model trained based on FFNN achieves the best performance. The five-fold cross-validation results of these two best-performing models on the training set were compared in Fig. 4 which shows that the fine-tuned Promoter-BERT model scores higher than the model based on extracted embeddings on 4 evaluation metrics, with AUROC, AUPRC, MCC, and ACC improving by 0.003, 0.009, 0.002, and 0.003, respectively. Therefore, the end-to-end fine-tuned Promoter-BERT model was selected as our final model for predicting 5mC sites, which is named BERT-5mC.

## DISCUSSION

### Comparison with other state-of-the-art models

We compared our model to four others that were developed recently: BiLSTM-5mC, 5mC_Pred, iPromoter-5mC and DGA-5mC. As the dataset is the same for all models, the comparison is fair. Table 3 shows the five-fold cross-validation results of all the models on the training set, and Table 4 shows the predictive results on the independent test set. Because the AUPRC values were not reported for the other models, only five evaluation

**Table 4 Performance comparison between different models on the independent testing dataset.** The highest value in each column is shown in bold.

| Method | Sen | Spe | Acc | MCC | AUROC |
|---|---|---|---|---|---|
| iPromoter-5mC | 0.878 | 0.904 | 0.902 | 0.577 | 0.957 |
| 5mC_Pred | 0.895 | 0.920 | 0.918 | 0.625 | 0.962 |
| BiLSTM-5mC | 0.866 | 0.937 | 0.930 | 0.638 | 0.964 |
| DGA-5mC | **0.902** | 0.927 | 0.925 | 0.646 | 0.964 |
| BERT-5mC | 0.872 | **0.938** | **0.933** | **0.656** | **0.966** |

metrics (Sen, Spe, ACC, MCC, and AUROC) are compared, with the highest scores highlighted in bold.

As shown in Table 3, on the training dataset, our model, BERT-5mC, achieves the highest ACC, MCC, and AUROC of 0.932, 0.653, and 0.966, respectively. The model DGA-5mC achieves the highest sensitivity of 0.904. The model BiLSTM-5mC achieves the highest specificity of 0.940, which is comparable to our model. In addition, Table 4 shows the predictive results on the independent test set, which indicate that our model achieves the highest Specificity, ACC, MCC, and AUROC of 0.938, 0.933, 0.656, and 0.966, respectively. The model DGA-5mC achieves the highest sensitivity of 0.902. According to these results, our model, BERT-5mC, outperforms the other state-of-the-art model for predicting 5mC sites of promoters. Based on the ACC values on the training and the independent test sets, a one-side two-sample t-test shows that our model is significantly better than BiLSTM-5mC with a $P$-value of 0.019. In summary, our model BERT-5mC can serve as a powerful tool for predicting promoter 5mC sites genome-widely, which can extract key information about 5mC sites from nucleotide sequences and is expected to contribute to the large-scale annotation of 5mC sites.

We analyzed the possible reasons why the performance of our model is superior to the other models as follows: (1) The pre-trained BERT model can better represent the contextual information concealed in the DNA sequences than other encoding methods such as FastText which generates context-free embeddings or other handcrafted encoding methods. Because the residues in DNA sequences are not independent, the interactions between these residues determine the structure and the function of a DNA; (2) by using the domain-specific *corpus*, Promoter-BERT was pre-trained whose representability was further improved. Based on the good performance of our model, the pre-trained Promoter-BERT can be transferred to predict other kinds of modifications of RNA and DNA. The pre-trained model can even be improved based on other deep learning techniques such as contrastive learning.

## Analysis of BERT embedding features

In this section, we visualized the embedding features using the t-Distributed Stochastic Neighbor Embedding (t-SNE) dimensionality reduction technique. The same number of positive and negative samples in the test set are randomly selected for the experiments to obtain the t-SNE plots. As shown in Fig. 5, the red dots represent the positive samples and
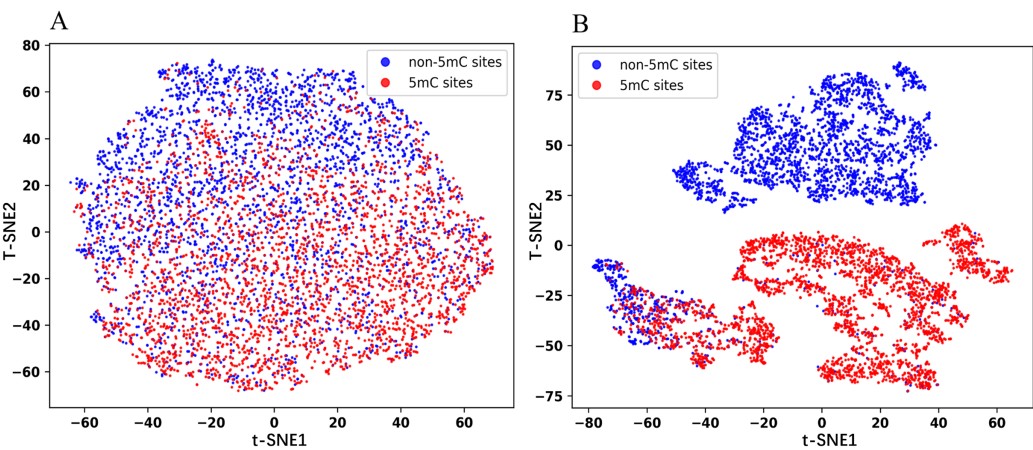

**Figure 5 t-SNE plots of the embeddings extracted from the Promoter-BERT model (A) and fine-tuned models (B) in this study.**

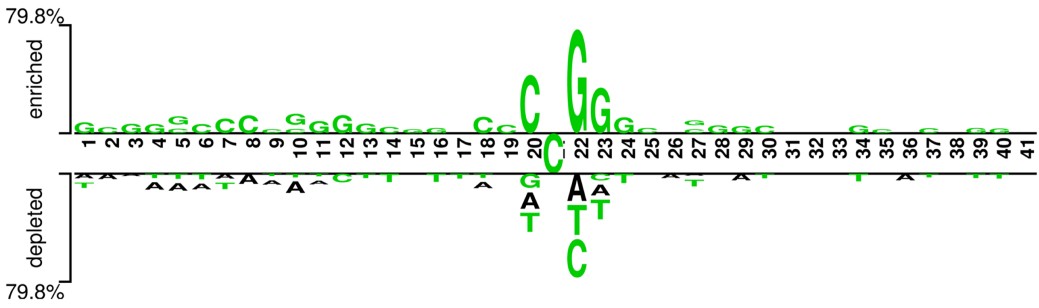

**Figure 6 Demonstration of nucleotide composition preferences between positive and negative samples in the benchmark datasets.**

the blue dots represent the negative samples. Figure 5A is the plot for the embedded features extracted from the pre-trained Promoter-BERT model, in which the positive samples are mostly distributed in the lower half of the image, while the negative samples are mostly distributed in the upper half of the image, but there is no clear boundary between the two groups. Figure 5B is the plot for the embedded features extracted from the fine-tuned Promoter-BERT model, in which there is a clear demarcation line between the positive and negative samples. This suggests that fine-tuning can help the BERT model learn to identify 5mC sites effectively, allowing a more accurate distinction between 5mC sites and non-5mC sites.

In addition, we analyzed if the attention weights of the samples are related to the nucleotide distribution difference between the positive and negative samples. Firstly, we plotted the nucleotide distribution difference between the positive and negative samples by using the Two Sample Logo (*Vacic, Iakoucheva & Radivojac, 2006*). As shown in Fig. 6, the distribution is significantly different at positions 19, 21, and 22. Then, we obtained the average attention weights of each word's (3-mer) position for positive samples and negative samples, respectively, based on the final encoder layer of the fine-tuned model. As shown in Fig. 7, the x axis indicates the positions, where each position corresponds to a

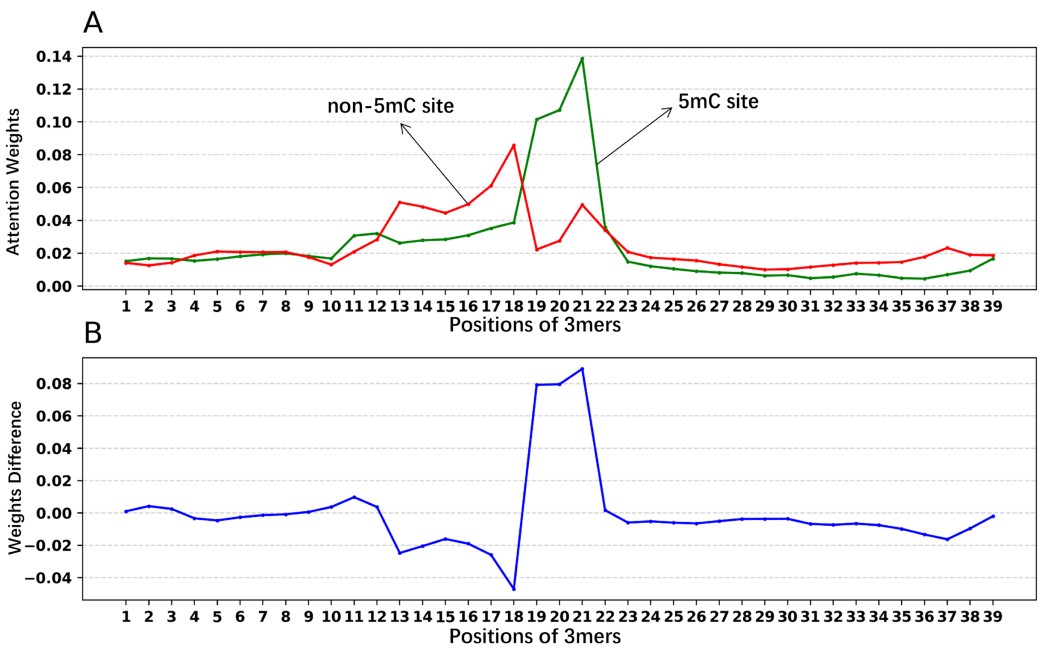

**Figure 7 Attention weights at different positions for positive and negative samples.** (A) Positional weights for positive and negative samples of the Promoter-BERT model. (B) The differences in the averaged position weights between the positive and negative samples were calculated for our Promoter-BERT model.

3-mer of nucleotide sequences, so that we obtained a total of 39 positions. The y axis represents the average attention weight (Fig. 7A) or the weight difference between the positive and negative samples (Fig. 7B).

Figure 7A shows that the attention in the fine-tuned model presents different values at different locations. This result implies that the fine-tuned model pays attention to key features at specific locations, thereby enhancing prediction performance. Figure 7B indicates that in the fine-tuned model, significant differences in attention weights between positive and negative samples occurred at positions 19, 20, 21, and 22, which is generally consistent with the locations of statistically significant nucleotide distribution differences (Fig. 6). This suggests that the BERT-5mC model utilizes attention weights to learn the nucleotide preference sites responsible for 5mC prediction, facilitating the model's ability to extract key information about the 5mC sites. Ultimately, nucleotide sequence patterns expressed by attention weights may be beneficial for exploring the relationship between that nucleotide sequence pattern and its associated biological function.

## Comparison with the fine-tuned model based on Ji et al.'s DNABERT model

DNABERT is a pre-trained DNA language model which was developed by *Ji et al. (2021)* based on human genome. To compare DNABERT with Promoter-BERT proposed in this study, we also fine-tuned DNABERT to predict 5mC sites, using the consistent training parameters with BERT-5mC. The cross-validation performance on the training dataset and the test results on the independent test are shown in the Fig. 8 and Table S6. In both

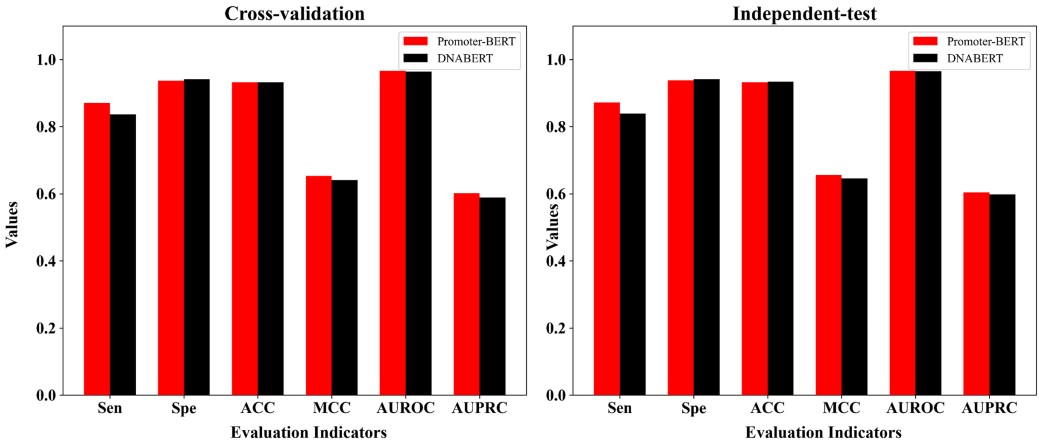

**Figure 8 Performance comparison between the fine-tuned model of Promoter-BERT and the fine-tuned model of DNABERT.**

five-fold cross-validation and independent testing, the fine-tuned model based on Promoter-BERT achieves higher scores for AUROC, AUPRC, and MCC than that of the DNABERT model. This suggests that fine-tuning on Promoter-BERT model performs better than fine-tuning on the DNABERT model of *Ji et al. (2021)* in the application of 5mC site recognition.

# CONCLUSIONS

In this study, we pre-trained a domain-specific BERT model specifically for the human promoters. To build models for predicting 5mC sites of promoters, we employed two strategies, namely fine-tuning the pre-trained Promoter-BERT model and extracting embedding features from the Promoter-BERT model. By comparing the performance of different models, the model based on fine-tuning was selected as the final model, which is named as BERT-5mC. The results on the benchmark datasets demonstrate the superiority of our model compared with the existing state-of-the-art models. We believed that our model could be a promising computational tool for 5mC site annotation. Additionally, given the diversity of DNA modification types, we can further fine-tune our pre-trained model on other DNA modification site prediction tasks in the future works. To facilitate the using of our model, we have set up a web server for our model at http://5mc-pred. zhulab.org.cn.

## Funding

This work was supported by the National Natural Science Foundation of China (No. 21403002), the Young Wanjiang Scholar Program of Anhui Province, and the Research Program of Education Department of Anhui Province (YJS20210223, 2023AH050998). The funders had no role in study design, data collection and analysis, decision to publish, or preparation of the manuscript.

## Grant Disclosures

The following grant information was disclosed by the authors:
National Natural Science Foundation of China: 21403002.
Young Wanjiang Scholar Program of Anhui Province.
Research Program of Education Department of Anhui Province: YJS20210223, 2023AH050998.

## Competing Interests

The authors declare that they have no competing interests.

## Author Contributions

- Shuyu Wang performed the experiments, analyzed the data, prepared figures and/or tables, authored or reviewed drafts of the article, and approved the final draft.
- Yinbo Liu analyzed the data, prepared figures and/or tables, webserver deployment, and approved the final draft.
- Yufeng Liu analyzed the data, prepared figures and/or tables, authored or reviewed drafts of the article, and approved the final draft.
- Yong Zhang conceived and designed the experiments, authored or reviewed drafts of the article, and approved the final draft.
- Xiaolei Zhu conceived and designed the experiments, authored or reviewed drafts of the article, and approved the final draft.

## Data Availability

The data and code are available at GitHub and Zenodo:

- https://github.com/sgrwang/BERT-5mC.

- ShuyuWang. (2023). sgrwang/BERT-5mC: First release of BERT-5mC (v1.0.0). Zenodo. https://doi.org/10.5281/zenodo.10143217.

The data used in this study is from Zhang et al. (https://doi.org/10.3389/fcell.2020.00614).

## Supplemental Information

Supplemental information for this article can be found online at http://dx.doi.org/10.7717/peerj.16600#supplemental-information.

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
