# Peer review of "BERT-5mC: an interpretable model for predicting 5-methylcytosine sites of DNA based on BERT"

_PeerJ, doi:10.7717/peerj.16600_

## Round 0.1 · original submission · Major Revisions

Both reviewers have concerns regarding the dataset selection. Please consider revising the manuscript and explain the dataset selection criteria. Reviewer #2s concerns on several decisions (such as 0.3 threshold) without statistical explanations should be addressed. Also consider publishing the code along with the paper for greater reproducibility of the results. More work would be needed to address the reviewers concerns especially Reviewer #2. Please submit a revised version after addressing reviewers comments.

·

Basic reporting

At first, I would to mention the actual topic of the study – to build predictive model to locate a 5mC sites in the genome data. Introduction section is very brief and doesn’t cover some recent studies on the topic (5mC sites prediction) to compare with:
• Jin, J., Yu, Y., Wang, R. et al. iDNA-ABF: multi-scale deep biological language learning model for the interpretable prediction of DNA methylations. Genome Biol 23, 219 (2022). https://doi.org/10.1186/s13059-022-02780-1
• DGA-5mC: A 5-methylcytosine site prediction model based on an improved DenseNet and bidirectional GRU method https://doi.org/10.3934/mbe.2023428
• MuLan-Methyl - Multiple Transformer-based Language Models for Accurate DNA Methylation Prediction https://doi.org/10.1101/2023.01.04.522704

Experimental design

Some details of the study need to be clarified:
L87: The approaches such as whole-genome bisulfite sequencing and reduced-representation bisulfite sequencing should also be mentioned.
L141: the dataset is not balanced, how the train and test were prepared? The same dataset was used in paper about DGA-5mC, so the comparison could be interesting.
L179: why the number of nucleotides varies from 1 to 41? Does it mean the motif or sliding window length?
Figure 2: the graph usually shows the changing the variable(s) in time or dependence of another parameter. In this case histogram fits better.
Figure 5: the colors should be contrast, to separate the dots (sites) more easily.

Validity of the findings

The obtained results are good, but hard to reproduce. Tool has no requirements to use (Python and libraries version, dependencies, etc.)
I suggest to prepare the scripts set to reproduce authors’ results. The repository https://github.com/sgrwang/BERT-5mC contains instructions how to build and tune the model only. The detailed manual will help to use (an cite!) your tool. As an example from the same field I suggest https://github.com/GoekeLab/m6anet

Reviewer 2 ·

Basic reporting

The paper presents a comprehensive set of experiments and results; however, the overall writing style needs improvement to enhance readability. In addition, the figures could be enhanced to facilitate clearer comparisons.

Experimental design

The research question is clearly defined, and the paper demonstrates a novel approach to bridging the language model to genome sequencing modeling. However, the explanation of this bridge should be elaborated upon, rather than using word/sentence and k-mer/sequence interchangeably.

Regarding the dataset used, the paper mentions Zhang's dataset, but it is necessary to clarify how using this dataset ensures reliability and fairness. The author should provide an explanation of why Zhang's dataset is a suitable choice, addressing its characteristics and potential advantages over other datasets used in related works.

Furthermore, in the mentioned section (line 131-136), instead of focusing on how Zhang collected the data in their work, the author should provide more information about the dataset itself. It would be valuable to introduce the dataset, including details about the information it contains and its overall structure.

Additionally, the paper only evaluates three values (1, 3, and 5) for the selection of k value in k-mers. The author should clarify why these three values are sufficient to conclude the best k value, addressing potential limitations or trade-offs in the selection process.

Moreover, the paper lacks an explanation of how the dataset size and the embedding dimension affect the feature selection. It would be valuable to provide insights into how varying dataset sizes and embedding dimensions impact the feature selection process. Moreover, the rationale behind solely using the embedding of the "CLS" token should be clarified, as it is not currently evident.

Validity of the findings

The paper includes a comprehensive evaluation using five metrics. While the author mentions that MCC is more reasonable for unbalanced datasets, it is not clear why the other metrics are still necessary. The author should provide justification for including these additional metrics and explain their relevance.

Additionally, the paper mentions a decision threshold of 0.3 without providing an explanation for its selection. It would be beneficial to clarify what the decision threshold represents and why the specific value of 0.3 was chosen.

Additional comments

The paper claims an improvement of 0.003 in accuracy and other metrics. However, the significance of this improvement is not justified. It is essential to provide further analysis or statistical evidence to demonstrate the significance of the reported improvements, avoiding potential dataset-specific effects.

Regarding the figures, Figure 2 would benefit from using a bar plot instead of a line plot. Moreover, it is advisable to include clear x and y axis labels in all figures to enhance clarity and understanding.

---

## Round 0.2 · Minor Revisions

The reviewers agree that the manuscript in its revised format is ready with minor corrections. Please pay attention to Reviewer #1s additional comments on duplicated references etc. While the time and computational requirements would be a good addition, it is not essential for the revised manuscript. Once the minor changes are made, the manuscript will be ready for publication.

·

Basic reporting

pass

Experimental design

pass

Validity of the findings

I would to thank authors for the efforts to improve the manuscript, but it still requires some corrections.
The performance of the compared tools weren't measured. How many time and computational resources are required to run the software?

The comparison of different tools is made, but Discussion section lacks the interpretation. Why the new tool outperforms the others and how you see its future development?

Additional comments

The manuscript need some proofreading.

Figure 2: it's better to define the window size in the legend instead "fine_tun_1" "fine_tun_2" "fine_tun_3" captions

L410: "Figure 7B reveals that..." Figure cannot reveal anything, please rephrase

Some references are being duplicated:
Bhasin M, Zhang H, Reinherz EL, and Reche PA. 2005b. Prediction of methylated CpGs in DNA sequences using a support vector machine. FEBS Letters 579:4302-4308. 10.1016/j.febslet.2005.07.002

Ji YR, Zhou ZH, Liu H, and Davuluri RV. 2021b. DNABERT: pre-trained Bidirectional Encoder Representations from Transformers model for DNA-language in genome. Bioinformatics 37:2112-2120. 10.1093/bioinformatics/btab083

Reviewer 2 ·

Basic reporting

The authors addressed the concerns regarding readability and figures. The manuscript is clearer and well-organized. Literature is appropriately referenced with a good context provided. All terms and theorems are sufficiently defined.

Experimental design

The revised manuscript solidifies its place within the journal's scope, presenting clear original research. Enhancements have been made in clarifying the bridge between language models and genome sequencing modeling. Methodological descriptions are now detailed enough for potential replication.

Validity of the findings

Data presented appears sound and controlled. Additional statistical analysis fortifies the conclusions made, ensuring they are supported by the results presented. Conclusions remain well-connected to the original research question.

Additional comments

The authors have addressed prior concerns, resulting in a clearer manuscript. Explanations regarding dataset choice, threshold, and metrics enhance the paper's comprehensiveness. Revised figures improve clarity.

---

## Round 0.3 · Minor Revisions

The corrections were minor and I have checked the corrections. The manuscript is almost ready for publication.

The Section Editor has identified some minor corrections that are still needed.

> The authors need to explain what "BERT" is, defining it in the abstract and earlier in the introduction. It isn't until line 122 that the actual definition is provided, and it isn't until the methods that we found out the reference for this algorithm "BERT (Bidirectional Encoder Representations from Transformers), developed by Devlin et al. in 2018, is a deep two-way language representation model based on Transformer (Devlin et al. 2018)". So, I would at least mention this after the first mention of BERT in the intro along with defining what BERT is. And, in that sentence, you only need one reference to Devlin et al.

---

## Round 0.4 · accepted · Accept

BERT is now explained in the abstract and the Devlin et al paper is cited.